

# Spontaneous reoccurrence of "scooping", a wild tool-use behaviour, in naïve chimpanzees

Elisa Bandini and Claudio Tennie

School of Psychology, The University of Birmingham, Birmingham, United Kingdom
Department for Early Prehistory and Quaternary Ecology, The University of Tübingen, Tübingen, Germany

## ABSTRACT

Modern human technological culture depends on social learning. A widespread assumption for chimpanzee tool-use cultures is that they, too, are dependent on social learning. However, we provide evidence to suggest that individual learning, rather than social learning, is the driver behind determining the form of these behaviours within and across individuals. Low-fidelity social learning instead merely facilitates the reinnovation of these behaviours, and thus helps homogenise the behaviour across chimpanzees, creating the population-wide patterns observed in the wild (what here we call "socially mediated serial reinnovations"). This is the main prediction of the Zone of Latent Solutions (ZLS) hypothesis. This study directly tested the ZLS hypothesis on algae scooping, a wild chimpanzee tool-use behaviour. We provided naïve chimpanzees ($n = 14$, Mage = 31.33, SD = 10.09) with ecologically relevant materials of the wild behaviour but, crucially, without revealing any information on the behavioural form required to accomplish this task. This study found that naïve chimpanzees expressed the same behavioural form as their wild counterparts, suggesting that, as the ZLS theory predicts, individual learning is the driver behind the frequency of this behavioural form. As more behaviours are being found to be within chimpanzee's ZLS, this hypothesis now provides a parsimonious explanation for chimpanzee tool cultures.

## INTRODUCTION

A growing body of literature suggests that humans are not unique in their possession of culture (culture defined as: "behavioural variation that owes its existence at least in part to social learning processes", *Perry, 2006*). In fact, various taxonomic groups provide evidence for some such form of culture. For example, whales (*Cetacea*; *Rendell & Whitehead, 2001*), capuchin monkeys (*Cebus*; *Fragaszy et al., 2004*), New Caledonian crows (*Corvus*; *Weir & Kacelnik, 2006*), and great apes (*McGrew, 1998*; *Whiten et al., 1999*; *Van Schaik et al., 2003*) have all been suggested to have culture. Among all these, great apes, and in particular chimpanzees (*Pan troglodytes*), are often described as having the most extensive repertoire of cultural behaviours (*Sanz & Morgan, 2007*; *Whiten & Van Schaik, 2007*; *Koops, Visalberghi & Schaik, 2014*). As the challenge to understand how human culture evolved continues

Corresponding author
Elisa Bandini, elisa-bandini@hotmail.it

(one of the top 125 questions of our time, see *Science*, special 125th anniversary issue, 2005), particular focus has been placed on chimpanzee culture due to their close phylogenetic ties to modern humans and their potential for providing insight into the evolution of hominin material culture (*Tomasello, 1999*; *Whiten et al., 2009*; *Tennie et al., 2016*; *Koops, Furuichi & Hashimoto, 2015*).

The current widespread assumption is that chimpanzee tool-use culture is based on homologous social learning mechanisms to human culture (*Kummer & Goodall, 1985*; *Boesch, 1996*; *Whiten et al., 1999*; *Whiten et al., 2001*; *De Waal, 2001*). Human culture is most likely dependant on high-fidelity social learning mechanisms that transmit information faithfully enough to allow for the cumulative nature of our culture (the so-called ratchet effect; *Tomasello, 1999*; *Tennie, Call & Tomasello, 2009*). Although the exact mechanisms for this faithful transmission are still debated, imitation (including action copying) and special forms of teaching (imitation-based teaching) are often cited as requirements for the (seemingly) unique aspects of human culture (*Tomasello, Kruger & Ratner, 1993*; *Boyd & Richerson, 2005*; *Hoppitt & Laland, 2008*; *Tennie, Call & Tomasello, 2009*; *Dean et al., 2012*; *Kline, 2014*, although see also *Caldwell & Millen, 2009* and *Reindl et al., 2017* for recent evidence that imitation may not always be necessary for cumulative culture to emerge).

Although some claim for evidence of high-fidelity social learning in non-human great apes (*Whiten et al., 1996*; *De Waal, 2001*; *Whiten et al., 2009*; *Hopper, 2016*; *Musgrave et al., 2016*), the actual data for spontaneous high-fidelity social learning in enriched captive apes, (i.e., apes who live in social groups, but have not been trained by humans, intentionally or unintentionally (not enculturated), see *Henrich & Tennie, in press*), remains questionable (*Tennie, Call & Tomasello, 2009*).

Indeed, previous studies and observations have failed to show conclusive evidence of action-copying and (imitation-based) teaching in chimpanzees, leading chimpanzees to be categorised as emulators (reproducers of environmental results) rather than spontaneous imitators (where action copying would play a role; *Tomasello et al., 1987*; *Tennie, Call & Tomasello, 2006*; *Tennie, Call & Tomasello, 2009*; *Tennie, Call & Tomasello, 2012*; *Myowa-Yamakoshi & Matsuzawa, 2000*. Although see also: *Whiten et al., 2004*; *Hopper et al., 2007*; *Yamamoto, Humle & Tanaka, 2013*). The converse claim (which instead claims that non-human great apes can and do copy actions) stems mainly from the outcomes of so-called 'two-target' tests (e.g., *Whiten et al., 1996*; *Whiten, 1998*; *Whiten, Horner & Waal, 2005*; *Whiten & Mesoudi, 2008* for work with chimpanzees, *Custance et al., 2001* with orang-utans, and *Stoinski et al., 2001* with gorillas). However, using this very kind of task, it was also found that the demonstrations that allow for imitation (demonstrations which include action information, often called "full demonstrations") are not necessary for the observer apes to show the demonstrated target actions—demonstrations of pure environmental results have been shown to lead to copying in this task, too (*Hopper et al., 2007*). Furthermore, a variety of animals have been shown to be successful copiers in two-target tests (e.g., pigeons, *Lefebvre, 1986*; capuchins, *Custance, Whiten & Fredman, 1999*; *Dindo, Thierry & Whiten, 2008*; vervet monkeys, *Van de Waal et al., 2010*), and recently even reptiles have been found to copy targets in this kind of task

(*Kis, Huber & Wilkinson, 2016*). Thus, the two-target method does not seem to be measuring any special copying mechanisms (at least not for the presence of any otherwise rare ability that humans may share only with non-human great apes).

Due to the absence of convincing evidence for high-fidelity social learning in non-human great apes, it has been suggested that chimpanzee cultural behaviours must derive their form and stability from processes other than high-fidelity copying (or high-fidelity teaching; *Tennie, Call & Tomasello, 2009*; *Tennie, Call & Tomasello, 2006*; *Moore, 2012*). Thus, to truly test whether (any form of) social learning is indeed necessary for the expression of a behavioural form, how the behaviour first emerges needs to be examined. Yet, identifying the first natural occurrence of a behaviour (most often in the wild) is very difficult. Previous studies have instead attempted to seed behaviours in captive (and sometimes wild) groups to examine how the behaviour spreads across individuals. For example, a recent report on chimpanzee tool-use cultures identified how a behaviour (moss sponging) spread through a population once it naturally occurred (*Hobaiter et al., 2014*). Although studying how a behaviour emerges across individuals is important, examining the origins of the behaviour can provide valuable insights into the learning mechanisms that are required for its acquisition in an individual—including identifying when behaviour copying is not necessary. In the case of the recent report by Hobaiter and colleagues (*2014*), the authors argue that social transmission explains 85% of moss-sponging events in Budongo Forest (Uganda). Whilst we agree that social learning played a role in explaining these increases in *frequency* of the behaviour, the same data set also showed that moss sponging was independently innovated by at least two individuals in the population (namely the alpha male and alpha female, *Hobaiter et al., 2014*). The independent reinnovations of this behaviour demonstrate that individual learning fully accounts for the behavioural *form*, yet low-fidelity social learning facilitates its *frequency* across individuals (creating the observed population-wide patterns). In a recent follow-up study by *Lamon et al. (2017)*, the authors discuss the roles of individual and low-fidelity social learning in moss-sponging in Budongo Forest: 'Of course, each moss-sponger has to individually learn the behaviour, but in all likelihood, this was facilitated by the social influence exerted by other group members that acted as model'. We agree. In other words, high-fidelity copying does not appear to be strictly necessary to explain the spread of this behaviour across a population. And, given the growing literature on spontaneous innovations of wild-type behaviours[1] by naïve individuals across a growing number of animal species (e.g., nest-building in weaver birds (*T. c. cucullatus*; *Collias & Collias, 1964*); nut-cracking in capuchins (*Sapajus apella*; *Visalberghi, 1987*); hook-making in New Caledonian crows (*Corvus moneduloides*; *Weir, Chappell & Allritz, 2002*); and functional tool making in Hawaiian crows (*Corvus hawaiiensis*; *Rutz et al., 2016*); nettle-feeding in gorillas (*Gorilla beringei beringei*; *Tennie et al., 2008*); leaf-swallowing in chimpanzees (*Pan troglodytes*) and bonobos (*Pan paniscus*; *Menzel et al., 2013*); moss- sponging by (also wild) chimpanzees (*Pan troglodytes schweinfurthii*; *Hobaiter et al., 2014*); non-human great ape tool-use by human children (*Homo sapiens*; *Reindl et al., 2016*)), we argue that the *form* of tool-use behaviours in great apes appears across individuals fuelled by individual learning. In these cases, social learning is not required to explain the form of the underlying behaviour

[1] Here we use 'wild-type' behaviours to describe behaviours that are shown by wild non-human populations—including behaviours described as cultural in the literature (e.g., those described by *Whiten et al., 1999* and *Whiten et al., 2001*).

[2] Note that humans also have a ZLS, i.e., behavioural forms that do not require social learning—but humans can go beyond their ZLS by cumulative culture, which is enabled via their high-fidelity social learning (*Reindl et al., 2016*). Culture also interacts with human cognition itself, and so this process ultimately leads to ontogenetic cultural intelligence (*Tennie & Over, 2012*, and see also *Herrmann et al., 2007*; *Reindl et al., 2016*).

(which instead derives individually), but instead (low-fidelity) social learning facilitates the reappearance of the behaviour across individuals (what we would like to call: ''socially mediated serial reinnovations'' (SMSR)).

Thus, whilst human social learning transmits the form of a behaviour between individuals and consequently spreads the actual behaviour across individuals, other great apes (and possibly all other animals, too) may be forced to continuously 'reinvent the wheel' (metaphorically speaking) due to the forms of their behaviours being largely the products of independent individual learning.[2] If so, such innovations would have to be within the species' potential individual behavioural inventive repertoire, referred to as their 'zone of latent solutions' (ZLS, *Tennie, Call & Tomasello, 2009*). Social learning mechanisms (of a low-fidelity type) foster the release of the latent behaviour in others in the population—i.e., may be responsible for the *illusion* of a spread of a given latent solution—but these mechanisms are not necessary to explain the *behavioural form* that comes about across different individuals. The ZLS approach thus provides a more parsimonious explanation for chimpanzee behaviours, in which individual learning is hypothesised to be the main motor that drives the frequency of their tool-use behavioural forms as well as explaining the similarity in behavioural form across individuals, rather than assuming that social learning is necessary for the latter (and, especially, without the need to assume high-fidelity social learning mechanisms).

Still, the behavioural patterns in the wild demonstrate that occasionally even neighbouring communities differ in their behaviour (i.e., where genetic and environmental influences are kept to an absolute minimum—leading to the conclusion that these differences arose and are maintained by social learning; *Langergraber et al., 2010*). Thus, any theory that attempts to explain chimpanzee behaviour patterns must be able explain how such differences come about. The ZLS hypothesis provides the following explanation. Patterns such as these can be explained by social learning of a low-fidelity type increasing the frequencies of certain latent solutions once they are expressed in the first individual (or in several individuals at once, as was found in *Hobaiter et al., 2014*). In other words, such social learning processes must increase the likelihood of individual expressions of the latent solutions in question. Thus, once a given latent solution is expressed by the first individual(s), low-fidelity forms of social learning (which are widespread in the animal kingdom) then essentially act to homogenise the likelihood of individual expression of the behaviour within the affected community. In other words, non-human great apes would not be so much specialised in exceptional social learning mechanisms, but instead they would be specialised in increased levels (or: reach) of individual learning. The ZLS hypothesis therefore offers an, *at base*, individual learning account for the *form* of behaviours, with low-fidelity social learning acting as a facilitator for the innovation of behavioural forms across connected individuals. The result is the creation of between-population patterns of chimpanzee tool-use behaviours, i.e., what we set out to explain (e.g., population A might show tool-use X, while population B might not or population A might show variant A and population variant B of a tool use behaviour but where both variants are latent solutions). Thus, similarly to accounts that favour a major role for (high-fidelity) social learning, social learning is still required to explain the patterns of at least some behaviours seen

across populations—but it would be of a low-fidelity type (One that does not transmit the behaviours themselves).

The only way to ascertain whether chimpanzee tool-use is indeed best accounted for by a latent solutions approach is to directly test whether these behaviours can be expressed by naïve individuals (a direct prediction following from the ZLS hypothesis; *Tennie, Call & Tomasello, 2009*). The alternative approach, where high-fidelity social learning transmits the behavioural form, would instead predict that these forms cannot be spontaneously shown by individuals that are unconnected to the culture that keeps them in place (and form). In such tests, subjects are considered naïve if they are in this sense unconnected, i.e., they have never been trained in and/or have never seen the behaviour before. To ensure ecological validity, subjects should be so-called enriched captive apes (see *Henrich & Tennie, in press*). Subjects are then provided with the necessary raw material and motivation (e.g., food baits) to develop the target behavioural form (this is a latent solutions (LS) test; *Tennie, Call & Tomasello, 2009*). If the naïve subjects develop the target form, this demonstrates that social learning (of either low-fidelity or high-fidelity type) is not necessary for explaining the tested behavioural form (and it becomes unparsimoneous to assume that social learning is responsible for the form in the wild).

Data collected from LS studies can then be generalised to a species-level through one of the two ZLS standards (which we introduce here): the 'single-case ZLS standard' and the 'double-case ZLS standard'. The two standards reflect the varying relative complexity of animal tool-use behaviours. For relatively complex behaviours, such as chimpanzee nut cracking (which requires a specific technique preformed in a predetermined order and several objects in conjunction, *Boesch et al., 1994*), it is very unlikely that the behaviour is *ever* shown by pure chance (e.g., during display). Thus, for relatively more complex behaviours we only require a single demonstration of the behaviour in LS tests to conclude that the behavioural form is within the species' ZLS (i.e., the single-case ZLS standard). Relatively less complex behaviours, such as chimpanzee stick-use, have a slightly higher chance (though still low) of being demonstrated through chance alone. Therefore, for relatively less complex behaviours, we propose that *two* individuals must demonstrate the behavioural form independently from one another for it to be concluded to be within the species' ZLS (i.e., the double-case ZLS standard).

To test the latent solutions hypothesis we provided naïve captive chimpanzees with all the materials necessary to execute the behavioural form underlying algae scooping behaviour of wild chimpanzees (which we operationalize here as 'scooping', see section below). As is necessary for a latent solution test, we tested the chimpanzees without presenting them first with demonstrations of the target behaviour. Thus, we were able to isolate the roles of learning mechanisms, allowing us to examine whether social learning is *necessary* for this behavioural form to emerge in chimpanzees. If a tool-use behaviour does *rely* on social transmission (i.e., where the actual form of the behaviour is, and must be, socially transmitted—as is the case in modern human culture), then it should never occur in circumstances in which social learning is not possible: it should therefore never re-occur in a Latent Solution Test. If instead it is a latent solution, it *should* re-occur under such conditions (*Tennie, Call & Tomasello, 2009*). Due to the target behaviour being among the

relatively less complex behaviours of chimpanzees (a variant of stick-use), we applied the double-case ZLS standard, and required at least two individuals to independently show the behaviour for it to be classified a latent solution.

## Scooping

Algae scooping (not to be confused with 'algae fishing'; *Boesch et al., 2016*), is a behaviour observed in wild chimpanzees in Bossou, Guinea. The behaviour involves feeding on aquatic algae using herbaceous tools (*Humle, Yamakoshi & Matsuzawa, 2011*). These chimpanzees use tools to feed on *Spirogyra sp.*, a common form of algae in Bossou that often covers the surface of ponds, streams and lakes (*Humle, Yamakoshi & Matsuzawa, 2011*). Although algae scooping has also been described elsewhere (*Sakamaki, 1998*; *Devos, Gatti & Levrero, 2002*), Humle and colleagues (*2011*) provide the only description of the actual form of the behaviour. The authors (*Humle, Yamakoshi & Matsuzawa, 2011*) divided algae scooping in wild chimpanzees into six steps[3] : (1) select a stalk or stick, (2) detach it from the branch or bush, (3) modify its length, (4) remove the leaves, (5) insert it into the water and (6) *scoop the algae*, using a 'gentle swivelling action of the wrist' (*Humle, Yamakoshi & Matsuzawa, 2011*). Our study focused on the behavioural form of *scooping* and the accompanying actions (steps 1, 5 and especially 6). We concentrated on scooping because the selection, procurement and modification of sticks (steps 1–3) are already known to be widespread behaviours in chimpanzees, strongly suggesting that they can be individually innovated (see *Whiten et al., 1999*; *Gruber, Clay & Zuberbuhler, 2010* for reviews of tool-use in wild and captive apes). Likewise, we were not interested in how chimpanzees might learn that algae are edible or where they can be found. While such learning can also be, and presumably often is, socially mediated in chimpanzees (e.g., via local and/or stimulus enhancement, see description of social learning terms in *Whiten et al., 2004*), this kind of information (what and where) does not require the copying of behavioural form from other individuals. Thus, the question of how individuals learn what exactly to do *at* the location or *with* the new type of food would remain unanswered. Consequently, when examining whether high or low-fidelity social learning mechanisms are required for animal tool-use behaviour to emerge, logically, the experimental focus must be on the *behaviour* (the actions) itself. Here we focused on examining the necessary learning mechanisms behind the scooping tool-use actions (identifying the need for a stick, inserting the stick and using it to scoop by applying a 'swivelling action of the wrist') by testing whether they would reappear spontaneously in naïve chimpanzees without the aid of social learning.

To recreate the need for a scooping action we provided chimpanzees with floating, elongated bread-crusts out of immediate reach—thus affording a swivelling action with a stick to retrieve the food. Crucially, what "algae scooping" and "bread scooping" have in common is that they require an appropriate stick tool and the target scooping action to retrieve items from a water surface. We tested the ZLS hypothesis (*Tennie, Call & Tomasello, 2009*) on scooping behaviour by providing two groups of naïve, captive chimpanzees housed in a zoo in the United Kingdom with all the ecological requirements and motivation for this behaviour to emerge (appealing floating food that could only be retrieved using sticks in a scooping manner). If at least two of these scooping-naïve

[3]Note that whenever a behaviour is divided into steps, one necessarily has to make, to a certain degree, subjective decisions (e.g., on a coarse level, would one include the way from, say, the night nest to the algae as a separate step in the sequence? On a fine level, should one count the movement of single finger digits? (*Byrne, Corp & Byrne, 2001*)).

chimpanzees spontaneously used sticks to retrieve the floating food with actions similar to the one used by wild chimpanzees, then this would strongly suggest scooping as being a behaviour within chimpanzees' ZLS (following the double-case ZLS standard, see above). To the best of our knowledge, no previous latent solution test has been carried out on the origins of scooping behaviour in chimpanzees or any other non-human great ape.

## MATERIALS AND METHODS

### Terminology

Throughout this manuscript we mention the 'reinnovation' or 'innovation' of wild tool-use behaviours in chimpanzees. We use the term 'reinnovation' when the specific actions (such as 'swivelling' the wrist to scoop algae: *Humle, Yamakoshi & Matsuzawa, 2011*) recorded in a wild-type behaviour are observed spontaneously in naïve subjects. Here we follow the definition of innovation provided by *Reader & Laland (2003)*, in which innovation is: 'a process that results in new or modified learned behaviour and that introduces novel behavioural variants into a population's repertoire'. Crucially, the authors clarify that 'population repertoire is not meant to imply that all individuals in a population will necessarily acquire the novel behaviour, but rather that at least one individual in the population will behave in a manner not previously seen' (*Reader & Laland, 2003*). Thus, latent solutions can be described as innovations according to this definition.

Our focus lies on examining whether the form of these innovations in non-human animals derives via non-social processes, and to emphasise the hypothesised individual learning aspect of innovations, we only refer to the very first description[4] of a behaviour as an 'innovation' but we prefer to call to all subsequent re-occurrences of the same behaviour as reinnovations (e.g., a behaviour is counted as a reinnovation when a similar form of the behaviour appears in unconnected, naïve individuals (either in captivity—or in the wild (namely when the behaviour is also found in culturally unconnected wild populations)).

[4]This description can come from wild or captive data—but usually comes from the wild.

### Subjects

Fourteen captive chimpanzees, ranging from seven to 49 years of age ($M_{age}$ = 31.33, SD = 10.09), based in a zoo in the United Kingdom took part in this study. All the chimpanzees are housed in social groups and have access to two indoor enclosures and two outdoor enclosures (with observational windows for visitors) and two indoor management areas, which are out of view of visitors. Throughout the enclosure the subjects have access to enrichment apparatuses such as climbing ropes and hanging feeders and are regularly provided with other enrichment devices. Subjects are never deprived of food or water, and continued with their regular feeding routine throughout this study. All subjects participated voluntarily in this study.

The chimpanzees were housed in two groups. In Group 1, seven out of the nine chimpanzees were born and raised in captivity (three males and six females, mean age: 27.7 years). In Group 2, four out of five chimpanzees were born and raised in captivity (two males and three females, mean age: 30.8 years (see Tables S1 and S2 tables for more information)). Wild born individuals were originally from the Democratic Republic of Congo or of unknown origins, whilst the majority of the captive born individuals were

born at the testing institution. Owing to zoo management requirements, it was not possible to test each individual separately; so they were tested in their normal group settings. The groups are kept separate, and no observation between the two groups was possible during testing. The testing was carried out in their respective communal management areas, and no individual was excluded. This project was reviewed and approved by the University of Birmingham AWERB committee (reference UOB 31213) and by the host zoo following guidelines provided by the SSSMZP, EAZA, BIAZA and WAZA on animal welfare and research in zoological institutions. This study adhered to legal requirements of the UK, where the research was carried out, and adhered to the ASP principles for the Ethical Treatment of Primates.

In order to fully isolate the roles of social and individual learning in a given target behaviour, the subjects must be naïve prior to testing. To test for this, all the keepers were interviewed separately in order to assess whether the chimpanzees had any previous experience with similar tasks, behaviours or materials. We asked for a detailed description of any spontaneous tool-use they may have seen and all past research and enrichment exercises the subjects had participated in that might have been similar to the one presented here (see Table S3 for a summary of the subject's tool-use experience). The keepers independently confirmed that none of the chimpanzees in this study had previously been exposed to any tasks, behaviours or materials similar to the one provided in our current study. The keepers reported that the chimpanzees did have access to sticks before our study, but as our focus was not on general stick use (which is already known to be widespread in great apes and thus reinnovated multiple times; *Whiten et al., 1999*; *Whiten et al., 2001*) previous contact with sticks did not present a problem to our study. Crucially, the keepers confirmed that the tested subjects were naïve to the problem of having to retrieve out-of-reach food and to the scooping action. Thus, it is highly unlikely that the subjects in this study, despite having had access to sticks, had previous experience with the problem of retrieving food from a body of water through the use of sticks (there are no water surfaces in the enclosure). Furthermore, the keepers also confirmed, through a questionnaire and follow-up interviews, that the chimpanzees did not have any experience with the 'swivelling' action required for the scooping behaviour seen in the wild (*Humle, Yamakoshi & Matsuzawa, 2011*). Although the ideal conditions would involve testing a group of chimpanzees raised in a fully controlled environment, these conditions do not exist to the best of our knowledge, (and would, in any case, lead to ethical problems). Therefore, the best available option involves testing captive chimpanzees whose previous experiences can be confidently accounted for (as we did here).

## Procedure

A square plastic container (16 cm × 66 cm × 20 cm) was placed outside the enclosure's mesh and filled with room-temperature water. Three bamboo sticks, modelled on the sticks collected in the field (*Humle, Yamakoshi & Matsuzawa, 2011*) in Bossou (min. 35 cm and max. 98 cm long, mean: 66.5 cm- diameter min. 5 mm, max 30 mm, mean: 17.5 mm), were placed around the enclosure prior to the chimpanzees entering the management area (again, given our focus on scooping actions, the provision of detached sticks presented no
problem to our study design). Prior to testing, the food (bread) was left to harden for a week so that it would float on the top of the water. The bread was cut into 'half-moon' shapes, to allow for it to be retrieved using a scooping action, similarly to algae in the wild. Three pieces of prepared bread pieces (half-moons) per testing session were placed simultaneously in the water container right before testing began. See Fig. 1 for the experimental set up.

Testing began at around 12.30 pm each day. Once the chimpanzees were allowed into the management area, a ten-minute testing period commenced. Sessions were video recorded on a Sony HDR-CX330E handycam. The test was live coded by E1 (EB) and filmed by E2 (FR). All chimpanzees then had potential access to the apparatus. Each group was tested three times: twice on consecutive days, and then a third time after 28 days. It was live coded whether the subjects used a tool to retrieve the food; if they used a scooping technique (following the description by the *Humle, Yamakoshi & Matsuzawa, 2011*, including the target swivelling wrist motion described in the original report) or a different technique; whether there were any instances of stick modification; how the stick was inserted into the water container; and whether the attempt was successful or not (an attempt was coded as successful if the individual managed to retrieve a piece of bread, including the smaller pieces that formed when the crusts started to disintegrate, and transport it to the mesh).

## RESULTS

### Reliability coding

To assess inter-observer reliability, a naïve individual—who was not familiar with the task or the hypothesis—coded from the videos all the same categories that had been lived coded. These categories were coded for each attempt in all six videos. The overall Cohen's Kappa was calculated (for a total of 164 instances): there was very good agreement between the two coders, $K = .870$.

Within the first ten minutes of testing (HO: 6 min 23 s and LO: 7 min 9 s), two females, HO (33 years, parent-reared and captive born at the testing institution) in Group 1 and LO (37 years, hand-reared and captive born at the testing institution) in Group 2, independently retrieved the floating food using stick tools and a scooping action (See Video S1 for video clip of individual HO scooping the bread). No other subjects showed these behaviours, but note that, (a) throughout the experiment, attempts to use the tools by other members of the group were actively discouraged by HO and LO, who dominated the testing apparatus. Thus, it is possible that other individuals might have used the scooping technique if they had been granted access to the apparatus. And (b) because the individuals could not be tested independently, data from individuals other than the first are generally un-interpretable with regard to our research question, as once one subject expresses the behaviour, other individuals can no longer be considered target-naïve. Thus, in a group setting, only the first occurrence per group counts in a latent solutions test, as social learning can no longer be logically excluded afterwards. Given the absence of scooping demonstrations for HO and LO, as well as their established scooping-naivety at test (see above), these two individuals could not have socially learnt the behaviour, demonstrating that both independently reinnovated it.

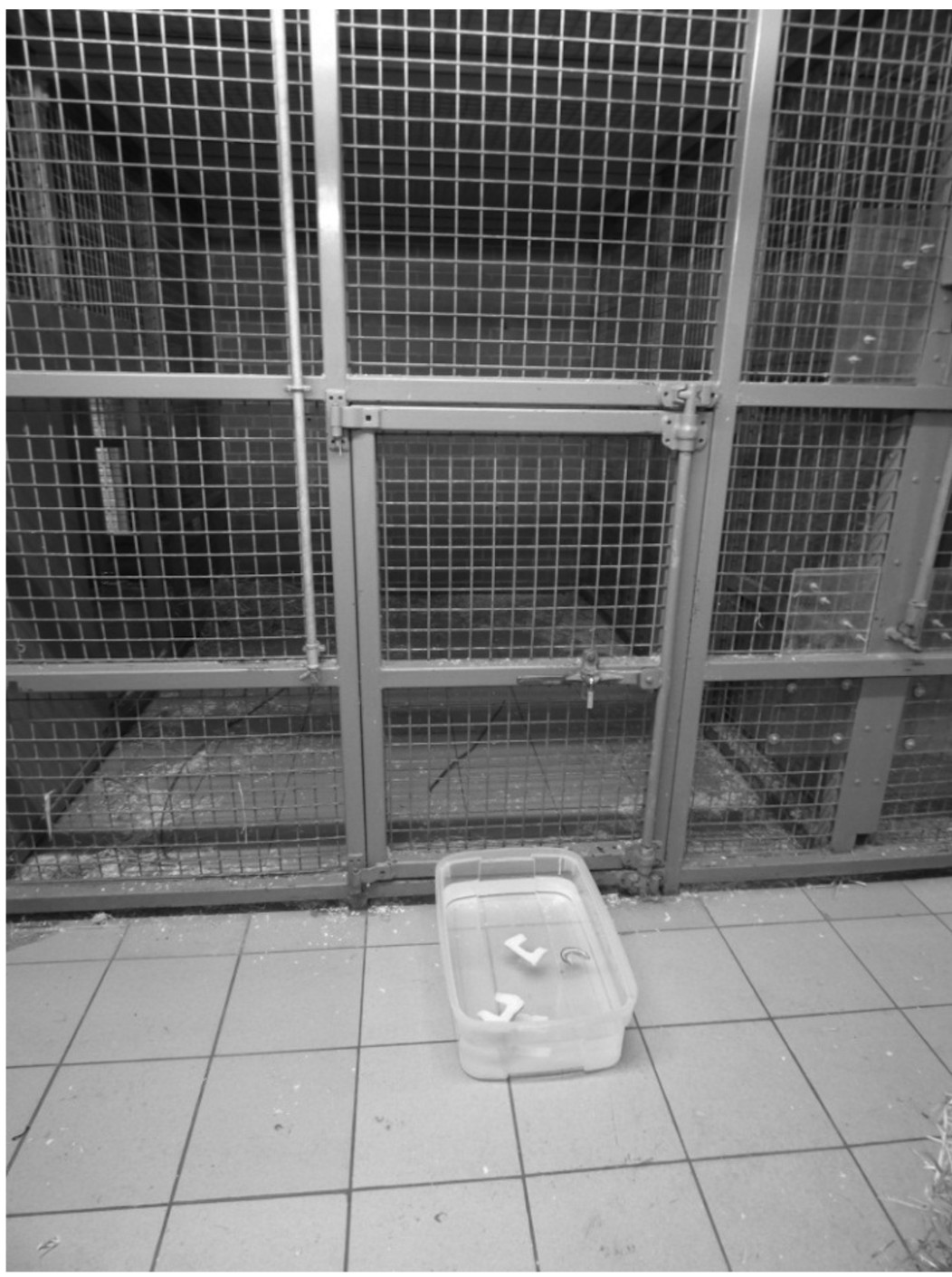

**Figure 1   Experimental set-up.** Container with bread crusts in the foreground and one of the sticks inside the enclosure (photograph by EB).

When scooping, HO and LO would insert the tool into the water above or close to the crusts and then gently rotate the wrist until the bread crust was wrapped around the stick. Once the bread crust was balanced on the tool, it was retracted towards the mesh. See Fig. 2 for an example of the scooping technique shown by HO.

The reinnovated scooping actions in our study were very similar to the wild scooping behaviour: the wild chimpanzees, as well as the two captive chimpanzees in the current study, scooped using "a gentle swivelling action of the wrist" (as described by *Humle, Yamakoshi & Matsuzawa, 2011* for wild chimpanzees).

It may still be of interest that, despite the focus of our study having been the scooping action, other steps of the wild algae scooping sequence were also recorded in our study. The basic sequences of the wild and our captive chimpanzees were very similar, although divergence existed between the order of some steps, with Bossou chimpanzees first modifying their sticks before inserting them into the water (most likely because they were detached directly from the tree or bush). Whilst the chimpanzees in our study were also observed to modify their sticks, they did so less frequently than their wild counterparts. Since the subjects in this study were provided with already detached sticks, they did not need to modify the length of the sticks; at least not as often as wild chimpanzees (and, as the sticks provided were already around the same length as that recorded in Bossou (*Humle, Yamakoshi & Matsuzawa, 2011*), further modification was not often necessary). A total of four instances of stick modification were recorded throughout our six testing sessions—that is, stick modification happened in 30% of all retrieval attempts (including unsuccessful ones). All modifications occurred after the sticks were first inserted into the water. In all these instances HO and LO used their fingers or teeth to break off a small piece of the stick, perhaps to make it into a more manageable length to retrieve the bread crusts that had floated too close to the mesh (all instances of stick modification occurred when the crusts were closest to the mesh, see Fig. S1 for stills on the stick modification method).

## Additional techniques

Due to slight differences in the overall physical setup between our experiment and the wild, we expected that the chimpanzees in our study would show additional behaviours. This was indeed the case, and both HO and LO were observed to occasionally make use of the sides of the water container to retrieve the bread crusts. The basic sequence of this 'side technique' was as follows: first, the stick was placed on the upper part of the bread crust, which was then pushed towards one of the sides of the bucket. Then, pressure was placed on the crust to slide it up the edge of the container and onto the rim. Once the bread was on the rim, it was pulled towards the mesh and retrieved with the fingers (see Fig. S2 for camera stills of this method). All side technique attempts to retrieve the bread pieces were also coded. In both subjects, the scooping technique was more commonly used than the side technique: in HO 68.9% (20/29) of attempts were with the scooping technique and 31.1% (9/29) of the attempts were with the side technique. In LO 61.8% (55/89) of the attempts to retrieve the bread crust were carried out using the scooping technique and 38.2% (34/89) were using the side technique.

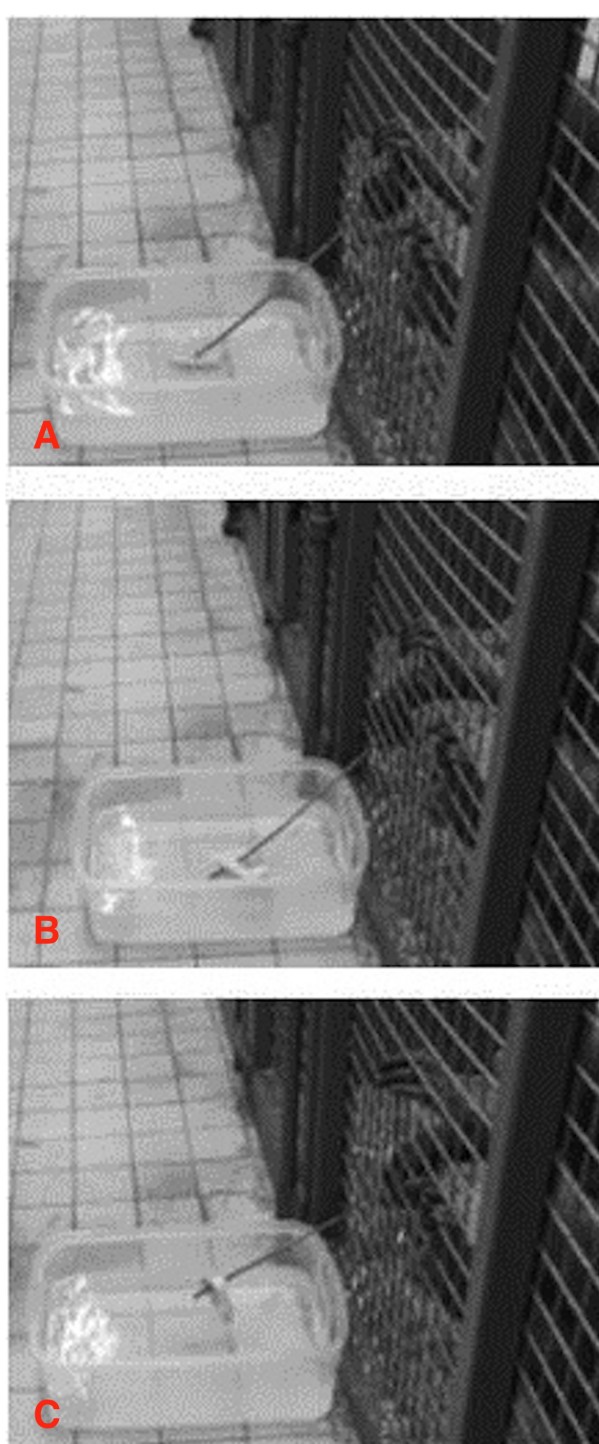

**Figure 2 Scooping sequence.** HO carrying out the scooping sequence. (A) HO inserts the stick under the bread, (B) using a 'swivelling' motion of the wrist, HO scoops up the bread (*Humle, Yamakoshi & Matsuzawa, 2011*) and (C) HO retrieves the bread (camera stills by EB).

**Table 1 Individual action variations.** Number of times each action variant seen in the wild was performed by captive chimpanzees (only clearly visible instances were coded, including instances in which the stick was manipulated and no attempt was made).

| Wild Behaviour (*Humle, Yamakoshi & Matsuzawa, 2011*) | HO/total | LO/total |
|---|---|---|
| Stick held between thumb and index finger | 22/45 | 31/44 |
| Stick held between middle and index finger | 23/45 | 13/44 |
| Direct mouth feeding | 8/21 | 0/12 |
| Use of fingers to feed | 13/21 | 12/12 |

## Success levels

In all three trials, both HO and LO retrieved all three pieces of bread crust (including small pieces which resulted from some disintegration of the bread crusts) within a maximum of six minutes. Mean retrieval time for each bread piece using the scooping technique in Group 1 (HO) was 4 s (SD = 1); in Group 2 (LO): 8 s (SD = 3; recorded from when the tool came in contact with the piece to when the individual started to feed). Mean retrieval time using the side technique in Group 1 (HO) was 20 s (SD = 12); in Group 2 (LO): 7 s (SD = 2).

## Individual variation in scooping technique

Individual variations in scooping technique were observed in the wild (*Humle, Yamakoshi & Matsuzawa, 2011*). Most frequently, Bossou individuals held the tool between the thumb and the index finger when scooping, but occasionally some gripped the tool between their middle and index fingers—although the exact number of times each variant occurred was not reported (*Humle, Yamakoshi & Matsuzawa, 2011*). Additionally, after scooping, some chimpanzees fed on the algae directly from the stick, whilst others, more rarely, gathered the algae off the stick with their fingers and then licked it off their hands. As in the wild, there were also individual differences between grips and feeding methods in our test subjects. To identify potential individual differences we coded all clear cases of finger positioning and feeding methodologies for HO and LO (instances were not coded if the video was not clear enough to identify grip or feeding method). Table 1 shows the frequencies of these variants between HO and LO.

As can be seen in Table 1, HO varied continuously between grips, and showed no preference for the middle and index grip whilst LO showed some preference for holding the stick between the thumb and index finger, similarly to Bossou chimpanzees. Furthermore, HO occasionally used the stick directly to feed, but preferred to use her fingers. LO only used her fingers to feed. Thus, overall, a comparable range of individual differences to wild chimpanzees were observed in this study.

## DISCUSSION

Our results demonstrate that the wild form of scooping behaviour re-appeared independently in two naïve chimpanzees (it was reinnovated twice). Thus, unlike human cumulative cultural behaviour, the observed patterns of scooping behaviour in the wild can be explained via Socially Mediated Serial Reinnovations (SMSR), rather than requiring

high-fidelity social learning mechanisms. As the scooping behaviour was independently reinnovated by two naïve chimpanzees, this fulfils the most conservative requirement for a latent solution (the double-case ZLS standard), and it strongly suggests that chimpanzees elsewhere also have the potential to produce this behaviour individually (though they may of course still be socially influenced in, e.g., where to feed and what to feed on when using this technique). Scooping behaviour is a latent solution in chimpanzees.

Given these findings, a latent solution account is not only probable for the first chimpanzee(s) who innovates the scooping behaviour in a particular group (e.g., by beginning to eat surface algae using a tool), but also for those who then "join in" due to low-fidelity social learning. The type of social learning used is most likely one that utilises each chimpanzee's ability to reinnovate the behaviour—but does not transmit the behavioural form itself (i.e., the social learning is not of high-fidelity type). Thus, our results strongly suggest that each individual chimpanzee is capable of reinnovating the behaviour independently, and that for those surrounded by others who already have expressed the behaviour, low-fidelity social learning mechanisms simply facilitate their own expression of this behaviour—increasing (and harmonizing) the frequency of individuals reinnovating the behaviour in the population (SMSR).

As a thought experiment, if we were to imagine all forms of social learning—including low-fidelity social learning—were completely absent from all chimpanzees, following the ZLS logic, behaviours such as scooping would still re-appear (though in many cases, rarely), given the right circumstances.[5] Indeed, scooping in the wild has also been reported outside the potential "cultural reach" of Bossou (*Humle, Yamakoshi & Matsuzawa, 2011*), namely in Odzala National Park, Congo (around 3,000 km apart; *Devos, Gatti & Levrero, 2002*). Why then, do we not see more populations engaged in algae (or other food) scooping? Perhaps this is due to local trade-offs between the necessity and the opportunity hypothesises (e.g., *Fox, Sitompul & Schaik, 1999*), a possible explanation for the fact that most wild innovations never "catch on" (*Nishida, Matsusaka & McGrew, 2009*), i.e., do not lead to SMSRs (more on this below).

This study provided evidence that chimpanzee scooping, a tool-use behaviour, is a latent solution (just like other (non-tool-use) great ape behaviours that have been tested following the Latent Solution Test methodology (*Tennie, Call & Tomasello, 2009*; *Tennie et al., 2008*; *Allritz, Tennie & Call, 2013*; *Menzel et al., 2013*; *Reindl et al., 2016*). In its current, strong formulation, the ZLS hypothesis makes a clear prediction: *every* wild-type non-human great ape behaviour should reappear in at least some subjects of the same species[6] who are naïve to the behaviour in question when tested in latent solutions test settings (*Tennie, Call & Tomasello, 2009*; *Henrich & Tennie, in press*). If this is the case, then human and chimpanzee cultures are ultimately founded on different underlying mechanisms.[7] Over time, this dissimilarity leads to very different downstream effects: a restriction to behaviours drawn from the individually-bounded "zone of latent solutions" in chimpanzees versus the open-endedness of cumulative culture in humans (*Tennie, Call & Tomasello, 2009*; although note that despite possessing extensive social learning abilities, human children are surprisingly poor innovators, e.g., *Beck et al., 2011*; *Nielsen, 2013* but see also *Reindl et al., 2016*; *Neldner, Mushin & Nielsen, 2017*).

[5]The likelihood of innovation is influenced by ecological (e.g., presence of algae) and behavioural ecological conditions (e.g., the nutritional need for algae in a given population—and these needs may differ between populations based on the effects of their ecology and their already adopted latent solutions). For more detail on the opportunity and necessity hypotheses see, e.g., *Fox, Sitompul & Schaik (1999)*.

[6]Though note that sometimes these behaviours also appear in different species, due to e.g., phylogenetically-shared parts of their ZLS.

[7]Humans have their own ZLS (*Vygotsky, 1978*; *Reindl et al., 2016*; *Reindl, Bandini & Tennie, in press*), but can and do copy the forms of behaviours outside their ZLS (*Tomasello, 1999*; *Henrich, 2015*).

We are not claiming that chimpanzee tool-use behavioural forms are genetic, in the sense that they have been individually directly selected for by natural selection. We do not envision a genetic structure that directly encodes scooping behaviour. Instead, apes have specialised in enhanced individual learning, i.e., in innovations—and, at least for chimpanzees (*Whiten et al., 1999*; *Whiten et al., 2001*) and orangutans (*Van Schaik et al., 2003*), this is already well expressed by their varied use of tools in the wild (the other great apes showcase these skills, too, but do this more so in captivity). The unspecialised, low-fidelity social learning mechanisms that apes use are piggybacking on these innovative powers (we hypothesize that ape cultures are based (perhaps in their entirety) on such socially mediated individual reinnovations). In this synergy between individual and social learning, apes do not seem to be very special—indeed, social and individual learning is highly correlated across the primate range (*Reader & Laland, 2001*). Yet, in their absolute levels of complexity they can reach in this way (e.g., see the case of nutcracking—but also the sheer number of different tool uses that are thus enabled), great apes are exceptional animals (alongside some bird species; e.g., *Weir & Kacelnik, 2006*; *Rutz et al., 2016*).

## Target scooping action

This study focused on the scooping action, the target behavioural form for which we examined the role of social versus individual learning in its emergence. Both wild (*Humle, Yamakoshi & Matsuzawa, 2011*) and naïve chimpanzees (this study) show this behavioural form (in particular, they rotate their wrist to wrap the food around the tool, before retracting it towards them). Our study suggests that this technique is rather easily reinnovated by individual chimpanzees, given (a) the latency with which they expressed the technique, (b) that two subjects did so and (c) that none of our successful test subjects had an opportunity to observe this behaviour previously or during testing. Thus, our data renders it parsimonious to assume that the scooping technique in the wild also arises on an individual level—as a latent solution.

Our conclusion is notwithstanding the fact that great apes in captivity have been shown to be generally more proficient and/or motivated to use tools than those living in the wild, a phenomenon known as 'the captivity effect' (*Tomasello & Call, 1997*; *Van Schaik, Deaner & Merrill, 1998*). The captivity effect does not impact our findings—or indeed any other latent solution experiment outcome—as the effect merely increases the likelihood of individual expression, but does not prescribe the behavioural form itself. To the best of our knowledge, the ZLS approach is best suited in providing an explanation for the similarities in behavioural forms that are observed across independent individuals—as for example in the present study.

## Individual differences

Individual differences in single actions during scooping behaviour observed in Bossou chimpanzees have seemingly been suggested as evidence for social learning: "Individual variations in the different algae-feeding techniques described here also should be further explored. The patterns of intracommunity patterns of algae-feeding techniques may correlate with observational learning [...] and thus purport a social learning mechanism

in their transmission" (*Humle, Yamakoshi & Matsuzawa, 2011*). However, comparable differences in action-level techniques were also found between our captive subjects—despite the fact that our subjects could not have observed the Bossou chimpanzees. The existence of these small individual differences shown by naïve chimpanzees in this study suggests that these differences are also a product of individual, rather than social, learning. In general, a more convincing argument for social learning in the wild would have been similarity of details of tool behaviour within a community but systematic differences between groups (including our study), unrelated to ecological and/or genetic differences. Currently the evidence for such variations in wild chimpanzees is limited (*Langergraber et al., 2010*), and even when such differences are observed (*Luncz & Boesch, 2014*), they do not reflect differences on the level of behavioural form. The observed differences can instead be explained through low-fidelity social learning mechanisms such as stimulus enhancement (for example in explaining the relative use of wood hammers versus stone hammers when nut cracking, as in *Luncz & Boesch, 2014*).

## Potential objections

Although only one chimpanzee in each group demonstrated the scooping behaviour targeted in this study (due to the apparatus being monopolised by these successful individuals) this is sufficient data to suggest that the behaviour is a latent solution for chimpanzees. Previously it was argued that it would suffice for only one individual to spontaneously show the behaviour for it to be considered within the species' ZLS and that even a single innovation would logically demonstrate that social learning is not necessary for it to occur (*Tennie, Call & Tomasello, 2009*). In our study, we observed the spontaneous reinnovation of scooping actions not only in one, but in two, independent chimpanzees, who never received any relevant demonstrations, training or experiences, thus fulfilling the even more stringent requirements for the double-case ZLS standard, which we propose for relatively less complex behaviours as the one tested here (see above).

In no way does our data negate a role of low-fidelity social learning in scooping, or any other chimpanzee behaviour when looking at the population level. Indeed, (low-fidelity) social learning mechanisms likely homogenise the likelihood of individual learning of many chimpanzee behaviours, and therefore (though not explaining the actual form of the behaviours in question) can play a decisive role in explaining the distribution of these behaviours as they are observed in ape populations. Once an individual has innovated the behaviour, several low-fidelity forms of social learning can help homogenise this behaviour across the population, by way of sustaining chain reactions, where each single reaction consists of an individual reinnovation (leading to the final "cultural pattern" observed in the wild; *Whiten et al., 1999*; *Whiten et al., 2001*). The only caveat to this domino-effect explanation is that not every latent solution will actually "catch on" once it is innovated in a given group (most innovations in wild chimpanzees do not, in fact, spread; *Nishida, Matsusaka & McGrew, 2009*). The main reason for innovations not spreading in the wild are not altogether clear yet, but may be related to the fact that wild apes are rather neophobic and thus de facto unlikely to adopt behavioural forms (contra the captivity effect for captive apes; *Forss et al., 2015*), and so are unlikely to reinnovate. In addition, perhaps meta-rules

(so-called social learning strategies) exist in apes for when to apply their low-fidelity social learning mechanisms, which might act against the usually observed type of wild innovators being influential (*Kendal et al., 2015*). And if, in addition, current claims for a majority influence in chimpanzees (*Luncz & Boesch, 2014*) can be further substantiated, then these effects could also proof detrimental to the uptake of latent solutions. Of course, latent solutions that are new to a wild group of apes sometimes do become population-wide behaviours after they are innovated (or at least occur lastingly across several individuals, see *Lamon et al., 2017*)—but this happens rarely and (currently, at least) unpredictably. Nevertheless, our alternative ZLS view explains the population differences we see across ape populations at least as well as the currently more widespread high-fidelity social learning account. In addition, the ZLS explanation stands alone in not requiring the additional assumption that apes are able to socially learn with high-fidelity (an ability that they may well lack, see 'Introduction').

Some objections on the results of this study may still remain. Firstly, some may claim that one cannot fully discount that the chimpanzees in this study saw demonstrations of the scooping action previous to our test. Given that apes are long-living animals, and not observed 24 h a day, the field has little hope to ever be able to negate such *ad hoc* claims (see also 'Introduction'). However, we must ask how likely it actually is that the chimpanzees were exposed to a similar behaviour before testing, given that the keepers all (independently) confirmed that subjects were naïve to scooping. This is especially the case here, where we detected scooping in *two* groups—thus, the behaviour would have had to remain unobserved by the keepers in not just one, but both groups of chimpanzees.

The chimpanzees in our study did have access to sticks before testing, which some might argue threatens their naivety to parts of the task. However, in this study we merely follow the accepted standard in field studies, in which, despite similarities in tools and actions, each chimpanzee behaviour is classified separately (see, for example: 'ant fish' and 'termite fish', both of which involve the same tool and action, but the different food sources being accessed is used to qualify them as separate behaviours; *Whiten et al., 2001*). The aim of our study was not to assess the tool-use abilities of completely stick-naïve chimpanzees, (also because it is practically impossible to find stick-tool-naïve chimpanzees in captivity), but rather to assess whether chimpanzees who are naïve to the scooping behavioural form, as described in the literature (*Whiten et al., 1999*; *Whiten et al., 2001*; *Humle, Yamakoshi & Matsuzawa, 2011*) would demonstrate the same behavioural form in the absence of social learning opportunities. Thus, whilst the chimpanzees in this study had experience with sticks, they were never faced with the problem of having to retrieve floating out-of-reach food inside a body of water. Importantly, the chimpanzees were naïve to the 'swivelling' wrist action required for the behaviour, i.e., to the key part of our target behavioural form (*Humle, Yamakoshi & Matsuzawa 2011*). Therefore, the subjects were naïve to the main aspect of the task—i.e., the target behaviour—making them ideal candidates to assess whether they would spontaneously solve the problem in a similar way to their wild counterparts.

Furthermore, it might also be objected that "extraordinary claims require extraordinary data". The extraordinary claim in our case might be argued to be that scooping represents

a latent solution, not necessitating social learning to emerge across individuals. We agree with the notion that extraordinary claims do require extraordinary evidence. However, 'extraordinary' evidence supporting the latent solution hypothesis in apes has already been provided (see also list above): naïve chimpanzees and bonobos independently reinnovated leaf swallowing behaviour (*Huffman & Hirata, 2004*; *Menzel et al., 2013*) despite never observing the behaviour in others before testing (and one can be 100% sure of this, as the leaf swallowing behavioural form occurs within the mouth area, and is therefore entirely concealed to observers). Therefore, our claim that a given great ape behaviour represents a latent solution must no longer be regarded an extraordinary claim. Following the same logic as these previous studies, our study—demonstrating two independent cases of reinnovation across two separate groups—therefore provides conclusive evidence supporting the notion that scooping lies within chimpanzees' ZLS.

## Conclusion

This research extends and supports previous work on the ZLS in great apes (*Tennie, Call & Tomasello, 2009*; *Tennie et al., 2008*; *Allritz, Tennie & Call, 2013*; *Menzel et al., 2013*; *Reindl et al., 2016*) which also found that other wild type ape behaviours develop spontaneously in naïve individuals and do not depend on social transmission, yet ours is the first study (to the best of our knowledge), to apply the latent solution logic explicitly to a chimpanzee tool-use behaviour. Examining tool-use behaviours is especially relevant for the study of non-human great ape cognition and evolution and also for understanding the evolution of human material culture. Understanding whether chimpanzee tool-use is fuelled mainly by individual learning or if it has to rely on social transmission can aid in the reconstruction of the evolution of hominin tools, which we believe may also have been characterised (at least for a long time) by individual reinnovations sensu the ZLS hypothesis (*Tennie et al., 2016*; *Tennie et al., 2017*).

Our study also highlights the importance of re-evaluating chimpanzee cultures in the light of latent solutions (*Tennie, Call & Tomasello, 2009*). The classic method of exclusion (*Whiten et al., 1999*; *Whiten et al., 2001*; *Van Schaik et al., 2003*; *Robbins et al., 2016*), which detects behaviour patterns across wild populations has many commendable points and has sparked a flurry of research into animal culture. However, it is likely that many or all of these behaviour patterns come about via a combination of several factors, such as genetics, ecological and cultural factors (*Laland & Janik, 2006*; and in our view, the latter consisting of low-fidelity social learning that is ultimately fuelled by individual learning). It is important to delve further into the underlying mechanisms *of each behaviour* by submitting them to latent solution testing as in the current study, especially before assigning them cultural status in the modern human sense of the word. *Whiten (2000)* best embodied this approach when stating: "the nature of the cognitive process of transmission matters in understanding what kinds of traditions, or cultures, really operate among nonhuman primates". We could not agree more.

Our latent solution approach for tool-use is new to great apes—at least when applied to wild type behaviours—but has already been tested in other species (for example, with New Caledonian crows (*Kenward et al., 2005*) and woodpecker finches (*Tebbich et al., 2001*)).

Using the latent solution methodology—providing naïve individuals who have never had the opportunity to see or learn from others with the ecological set-up of materials and reward structures—will further aid in identifying the necessary underlying mechanisms and their relative roles in the expression of that behaviour. Following this process we can better understand what forms of culture exist in both human and non-human animals—and which factors are shared and which are not. In this study we found that scooping is within chimpanzees' zone of latent solutions and therefore is not indicative of high-fidelity social learning. With more research in the field following the latent solutions method, we predict that several other behaviours, including those that were previously believed to *require* social learning (e.g., Whiten et al., 1999; Whiten et al., 2001) may soon follow suit.

## ACKNOWLEDGEMENTS

The authors are very grateful to Twycross Zoo for their kind cooperation and help for this research, in particular: Charlotte Macdonald, Simon Childs, Zak Showell, Sharon Redrobe, as well as all the animal keepers involved. The authors also thank Flo Rocque for her help collecting the data, Alice Coombes and Clare Williams for coding, and Eva Reindl and Zanna Clay for their comments on earlier drafts of the manuscript. We are grateful to the anonymous reviewers for their thorough and helpful comments on earlier drafts of the manuscript.

### Funding

This research was supported by a NERC grant (RRAL18516). There was no additional external funding received for this study. The funders had no role in study design, data collection and analysis, decision to publish, or preparation of the manuscript.

### Grant Disclosures

The following grant information was disclosed by the authors:
NERC: RRAL18516.

### Competing Interests

The authors declare there are no competing interests.

### Author Contributions

- Elisa Bandini conceived and designed the experiments, performed the experiments, analyzed the data, contributed reagents/materials/analysis tools, wrote the paper, prepared figures and/or tables, reviewed drafts of the paper.
- Claudio Tennie conceived and designed the experiments, wrote the paper, reviewed drafts of the paper.

### Animal Ethics

The following information was supplied relating to ethical approvals (i.e., approving body and any reference numbers):

This project was reviewed and approved by the University of Birmingham AWERB committee (reference number UOB 31213) and by the host zoo following guidelines provided by the SSSMZP, EAZA, BIAZA and WAZA on animal welfare and research in zoological institutions. This study adhered to legal requirements of the UK, where the research was carried out, and adhered to the ASP principles for the Ethical Treatment of Primates.

## Supplemental Information

Supplemental information for this article can be found online at http://dx.doi.org/10.7717/peerj.3814#supplemental-information.

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
