# Peer review of "Spontaneous reoccurrence of “scooping”, a wild tool-use behaviour, in naïve chimpanzees"

_PeerJ, doi:10.7717/peerj.3814_

## Round 0.1 · original submission · Major Revisions

I now have 2 reviews in hand, but with decidedly different bottom lines. The "minor revisions" review (Reviewer #1) raises an important issue related to generalizing to the population from N=2 observations. The "major revisions" reviewer (#2) is apparently familiar with your paper from its submission to another journal and calls for many clarifications. This reviewer's concerns are manifold and largely persuasive in my opinion. You will need to respond in detail before your paper can be considered acceptable for publication in PEERJ, and I hope you will accept this challenge.

·

Basic reporting

It is clearly written and argued, and I have no further suggested improvements on the basic reporting.

Experimental design

This is an original and important paper that provides a new theoretical basis for tool behavior in primates (and other animals as well, I would say).

I can envision this study as a jumping off point for similar, intriguing research - as the authors note, more studies are needed to identify which behaviors are within great apes’ spontaneous capabilities.

Validity of the findings

This is a sound study, and I mainly had minor questions. However, per Lines 109-115: This seems a post-hoc proposition based on the research presented here. I would feel more comfortable seeing this in the Discussion section of the manuscript. I don’t think it would take away from the authors’ arguments, and it follows their findings, which are convincingly argued. That said, however, “at least two independent target-behaviour-naïve individuals” does seem subjective given the results of this study. Again, I am persuaded by the authors’ arguments, but ideally would they say that the majority of the population would exhibit such behavior (i.e., if logistics such as those preventing the other subjects in the current study from exhibiting such behavior were not in place)? On a population level, this would seem to be a better argument, especially as the authors state in the Abstract that individual-level behaviors can help achieve population-level patterns. Perhaps it is beyond the scope of this paper to ascertain what proportion of a population would exhibit such individual behaviors, but it is something that is at least relative to their theory.

Additional comments

Line 64 – as far as neglecting the origins of behavior, might this be more along the lines of a rarity in terms of observing novel behaviors? (rather than neglecting discussion of origins) That is my understanding of the records of novel tool use behaviors that appear and then spread (or not) among wild chimpanzees at least.
Line 76 – could the behaviors be listed here with the relevant references?
Line 77 – “…but are also spread” seems to contradict the following sentence – at least in the way I read it. I’m not sure if rewording would help clarify…
Line 80 – “socially mediated serial reinventions” is a mouthful (but clear and accurate) - If this is used often, I would note “(SMSR)” here, following ‘socially mediated serial reinventions’ and use the acronym throughout
Line 83 – Does ‘the cumulative nature of our culture’ necessarily include active teaching? If this is implied, I would qualify this phrasing or somehow further elaborate. If this statement is just a broad one, I’m not sure how helpful it is to the reader as it is written – but perhaps I’m not following the connection between our cumulative nature and specifics of the ratchet effect (which I am familiar with)
Line 88- do humans also have ZLS?
Line 92 – should “spontaneous teaching” be “spontaneous active teaching”?
Line 93- could the authors references these findings again here (per “see above”)
Line 100- perhaps “can then spread”
Line 114 – could the authors clarify what they mean by enculturated?
Line 120 – could the authors clarify what they mean by unexceptional?
Lines 123-129 – per weaknesses of previous studies…I’m wondering if this might be better suited to the Discussion? Also, perhaps the authors should explicitly state how their study is providing baseline data – I’m not following their argument to the point at which I think they would like the reader to, given the pointed nature of this section…
Line 158 – change “decided to divide” to “divided”
Line 194 = spell out 14 at the beginning of the sentence (usually)
Line 197 = add comma between ‘areas’ and ‘which’
Line 206 – I’m assuming this is because of zoo management…? Could that be used instead of ‘local restrictions’?
Line 208 – can the sentence ending on this line be qualified with, “…during testing.”?
Lines 219-220 – this is interesting in that a challenge will be to actually find naïve chimpanzees for most tool tasks!

Reviewer 2 ·

Basic reporting

English is correct.

Literature covered is correct, but some refs are missing (see comments below).

Structure is fine.

Results: it's unclear whether the authors truly test what they are intending to test, as it is unclear whether chimps are truly 'naive' to the task of using a stick as a tool (see below).

Experimental design

Aims and scope: yes

Research question is well defined.

Investigation is performed ethically.

Methods are described with sufficient details to be replicated (see below for comments).

The major problem with the claims of the article is that it is absolutely necessary that the chimps are completely naive to the task. This can be easily tested with presenting them with an apparatus containing a valuable resource which is out of reach. Some populations in Africa have been found to be completely naive to stick use. If the chimps already know that sticks are potential tools, there is not much of a big cognitive jump to use a stick for a different task.

Validity of the findings

The novelty is not clear as the authors have already published similar claims before, albeit not on this particular behaviour.

There is not much to comment on the data as they are relatively simple.

Conclusions are speculative at this point because it is unclear whether the ZLS is tested by the current design (particularly because of the choice of the chimp subjects). I believe the discussion should be organized about chimp tool flexibility (which is demonstrated by this experiment). See below my comments.

Additional comments

I had already the opportunity to review this article for a different journal, providing the authors with a detailed review and comments to be addressed. Sadly, the authors did not address much of what I wrote so I am re-enclosing my former review, with the hope that this time, the authors will actually pay attention to my comments. Considering that the other reviewer for that journal had similar points to mine, I would assume these must be addressed if the paper is to be considered for acceptance. The lines have been re-adapted for convenience, and comments on changes by the authors or novel information have been added between ticks.

This article claims to bring a welcome test of the Zone of Latent Solutions (ZLS), applied to tool use. In general, I thought the article was quite well written, although the review of the literature appears too biased towards the hypothesis defended by the authors (that is, chimpanzee cultures result only from individual learning), rather than giving a balanced account of the current literature and debates and thus would benefit from being re-balanced. As such, this study follows on previous work by one of the authors, aiming to document whether ‘naïve’ apes can replicate behaviour observed in the wild, presented as ‘reinnovation’ or ‘reinventions’. First, I am glad to see the ZLS explained more than in the first study introducing the concept (Tennie et al. 2009). Particularly, I thought that highlighting two particular aspects of transmission, that is transmission of the form, and transmission of the ‘idea’, was quite convenient, especially to further the debate about the ZLS. However, I am unclear whether this particular study really tests the ZLS, rather than innovative abilities in chimpanzees. One major problem, I believe, is that the tested chimpanzees were already familiar with sticks (and bread). As such, their use of sticks to obtain floating bread is not surprising. The authors also make a whole case of why this result would be relevant to finding of ‘chimpanzee cultures’ in the wild. However, it is unclear how this study can have consequences on interpretations in the wild. Indeed this study really investigates flexibility in chimpanzees (how to make use of one’s knowledge to solve new problems), rather than the ZLS itself. In this respect, the abstract and main conclusion should be rewritten in this spirit, rather than focus on attacking the claims of chimpanzee culture in the wild, which the authors cannot address with this experiment. My comments follow the text:

Abstract

-L17-19: I find this statement quite confusing. First of all, down to the individual level, every behaviour learnt (including socially) depends on individual learning. Then, this learning may be influenced by social factors or not. As well characterized by Fragaszy & Visalberghi 2001, social learning is better defined as ‘socially-biased individual learning’ (see also Heyes, 2012, JCP for a more recent review). I think in the article, the authors make a better job of explaining what they mean. If I understood well, for them, ‘individual social learning in humans’ influences the form of the behaviour acquired. In contrast, in chimps (and other animals), at best, they get the idea of what they should do from others (thus socially) but have to re-invent it themselves. Why not explain it like that already here?

The organisation of the abstract is also slightly awkward, with much of devoted to explaining what the ZLS is, rather than actually introducing the study. The abstract may benefit from being rewritten more along the lines of standard abstracts.

Introduction

-L54-55: “ecologically appropriate chimpanzees” does not seem appropriate to me, particularly according to the definition given by the authors. Only wild chimpanzees are technically ‘ecologically valid’ chimps. At the same time, one can make a case that captive chimps’ environment is their ‘natural habitat’. Whether captive environment can however translate into valid representation of wild habitat is much more debatable. I think the authors should characterize better what they mean by this expression, but also acknowledge that captivity is a poor substitute to the wild.

-L56-60: I think this sentence should be rephrased, mentioning that some studies fail to find evidence of imitation, while others find evidence of imitation, leading to the conclusion that chimpanzees have a range of possible social learning mechanisms at hand, including imitation, but that they don’t rely on the latter all the time, and particularly not as much as humans. The way it is now is too biased toward the authors’ belief that there is no imitation in chimpanzees.

-L69-72: This is not correct. According to the authors of the article mentioned, social learning did play a major role (85-99% see abstract) both in the ‘spread’ and ‘the appearance of the behaviour of others’. Thus the sentence has to be modified to reflect this. Additionally, while the behaviour may have been rediscovered by one individual, this is because this individual picked up a moss-sponge herself, and the authors conclude that the most likely explanation for this transmission was some form of stimulus enhancement. This is social learning nonetheless. However, I believe this could go along the lines of what the authors discuss later on (in terms of low-level social learning mechanisms).

'Finally, the Zuberbühler group has produced a novel study that came recently (Lamon et al. Science Advances 2017), which brings further evidence of social learning. I thus encourage the authors to review their description of this particular set of studies if they want to make a point about social learning in chimpanzees in the wild.'

-L77-80: Although I believe this wording here (‘we argue….’) is clearer than in the abstract, does this statement not amount to the old Tomasellian argument about imitation vs the rest? Technically, only imitation would allow the transmission of the form of the behaviour (but again, it’s unclear whether imitation is really different than other social learning mechanisms, see Heyes 2015, Dev Science). In general, I would agree that it is interesting to discuss these two aspects of social learning, one that is more concerned with spreading the form of the behaviour, and the other one more concerned with spreading the idea, so to speak, but in that case, why would the authors not apply the same argument to humans? We could also say that the spread of the behaviour in humans results from socially mediated serial reinventions, all based on an incremental scale (based on each generation’s pool of knowledge).

-L92-95: I do not understand why the authors jump straight from imitation to individual learning (rather than, say, emulation, if that is what apes do according to them). As per their summary above, they were hinting at great apes being emulators rather than imitators. This can be debated, but let’s agree to that for the sake of the argument. Then, it is more parsimonious to assume that emulation is the main motor that drives the emergence of tool-use behaviours in apes, rather than individual learning. Again, we are still faced with the same problems: first social learning is, at its core, based on individual learning, but socially-biased. Second, the emergence, at the group level, also depends on innovation abilities of single animals. The authors need to make a better job of explaining their stance on these two topics (I have felt that the innovation side has been quite occluded in the intro).

'Compared to the former version, the authors have added a statement between brackets about the fact that there is a debate about imitation. While this bracket may be useful at a different place in the manuscript, this is not the point of the main sentence to which the bracket is connected and which states that “individual learning”, rather than any sort of social learning is actually at work in chimpanzees, which is not true!'

-L99-102: I’m unclear on how low social learning mechanisms may homogenize a given ‘form’ of behaviour so the authors need to develop this more. This is only possible if the pool of possibilities is itself quite limited, and thus cannot grant much diversity at the end. If not, one would expect to see more diversity in the use of one particular tool, equivalent to chance level to select a given option in the use of that tool.

-L103-104: I think the authors need to somehow come clean here. At the beginning of the intro, we are led to believe that the authors favour individual learning for the form, then social learning (albeit low level) for the spread, which could somehow characterize the ZLS. But now, it seems it is only the ZLS is only about individual learning. Which side are we supposed to take?
-L116-139: This paragraph should go higher in the introduction. Possibly before L105. In general, it could probably be beneficial to have some discussion about individual learning vs innovation too, and how they are taken into account in the ZLS framework (see my point above).

Methods

-L214-231: I believe the fact that the chimps already knew about stick use is actually a major drawback to the claim that the chimps are naïve in this study (see below). In fact, it would be indicated to summarize here or in the supplemental info how much knowledge the chimps had about stick use (e.g. termite mounds etc…), as per the carers’ statements.

'The authors still haven’t answered my comment about the kinds of ‘stick knowledge’ that was available to the chimpanzees. If the chimps were already used to use stick to recover food (e.g. in a termite mound or from a pot of some sort), there is no massive cognitive jump to use this knowledge to recover food from a body of water. I still would request a detailed description of the tool behaviour of the chimps prior to the experiment.'

Results

-L269-271: I am not clear with the statement about copying here. If we follow the argument of the authors in the introduction, then it’s all individual learning anyway, or simple stimulus enhancement, right? So why not analyze the behaviour of other individuals? This could possibly bring some data in favour of the authors if all individuals demonstrated a different technique to reach the bread.

-L274-275: Even though they discuss it at the beginning of the Methods, I am not clear why the authors use the word ‘re-invent’ here. They should rather use the word ‘innovate’. The chimps did not re-invent scooping, if we assumed they were naïve before, so they innovated it. I believe the authors need to address the difference between ‘innovating’ and ‘reinventing’ in the Methods. If not, they should use the word ‘innovate’ throughout the manuscript.

-L276-279: I am wondering about how much the fact that the chimps had to go through the mesh led them to be cautious and ‘gently rotate the wrist’. Basically, where their hand movements highly constrained?

-L285-292: Considering this statement, I do not think the authors can claim above that the behaviour was indistinguishable from the wild counterpart.

-L314-318: Usually success time (or any duration) is given +/- Standard Deviation.

Discussion

-L338-339: I don’t think any behaviour require ‘social learning’ to be ‘invented’ or ‘reinvented’. It requires innovation abilities by definition! So the authors are fighting a strawman here. The sentence should be modified.

-L341-343: I think one major caveat here is that the chimps did know about stick use before, and most especially, stick use to obtain food. It has been shown that wild chimpanzees have much difficulty to innovate stick use to obtain food when the behaviour is not present in their behavioural repertoire (see work by Gruber et al. 2011). Thus it is unclear whether the chimps in the present study needed in fact any learning at all (social or individual). They knew about sticks, possibly knew how to use sticks to get food (this absolutely needs to be cleared in a reviewed version of the article), they knew about bread, and thus, what they innovated at best was to use a stick to recover a food source. For the record, I am not saying that the wild chimps of Bossou did anything differently. They probably already knew stick use, and may have tasted the algae without using sticks at first. But this particular point needs to be addressed. It is fundamental to the argument of the authors.

-L352-356: Here there is a third definition/characteristic of ZLS: that all innovations are latent solutions. But then the whole ZLS argument should not be about social learning, but about innovation itself. But then the intro must be reshuffled to introduce discussion about innovation. In a nutshell, we are presented with several definitions/exemplifications of the ZLS. The authors need to pick one and stick to it!

-L372-375: I am unclear about why, evolutionarily, apes (I assume the authors exclude humans here?) would evolve enhanced individual learning, particularly considering that some populations of chimps and orangutans, and most populations of bonobos and gorillas are fairly limited in tool use. Also, why would have this disappeared in the human lineage in favour of social learning? In fact, humans are probably the best individual innovators, which is one founding block of cumulative culture (although children are, counter-intuitively, quite bad at it..).

-L379-382: The authors are still misinterpreting the chapter by Humle et al. The point of these authors relates to the similarity between individual techniques relating to proximity between specific individuals. It is not about “individual differences in single actions” but rather individual similarities. Therefore, the authors have to replace the word “differences” by the word “similarities”.

-L419: Again, it is unclear to me whether non-stick-using chimpanzees would actually be able to innovate this behaviour. This has been tested in the wild (see work by Gruber et al. 2009, 2011) and chimpanzees failed to “reinvent” the behaviour. I believe the authors should give reasons why they would think there was any innovation at all at play in their study. If anything (and taking into account the ‘slide’ behaviour) the chimps demonstrated flexibility in their behaviour and use of their old knowledge to solve a novel problem.

-L428-429: This is actually quite a ‘big’ caveat, because it seems that the ZLS would suggest ‘blind’ acquisition of the behaviour. That is, as long as some kind of low-level social learning mechanisms appear, the animal would not have controlled on whether to decide to do the behaviour or not. In a nutshell, it would necessary have to display the behaviour, which would then automatically ‘spread’.

-L432: I am quite uncomfortable with the use of ‘re’ something everywhere in the paper. While the authors mention it in the methods, they do not justify it. What is actually the case for it? ‘reinnovate’ to me, would amount to chimps having had the behaviour in a given population, which disappeared, and which subsequently re-appeared. This would qualify as ‘re-innovation’. But that zoo chimps innovate the same behaviour as displayed in the wild does not qualify as ‘re-innovation’.

'I’m still unclear about the use of re-invent, re-innovate, etc… Can’t the authors provide clear definitions for each of the words they use here?L189-192 are uninformative regarding this.'

-L441+: While I congratulate the authors for addressing potential criticisms, I feel like the most important one that has not been addressed is the fact that chimps all knew how to use sticks, irrelevantly of whether they saw or not the actual scooping behaviour. That is, they already knew that sticks are tools that can be used to recover food. Then there is not much needed to ‘rediscover’ the behaviour, under the form of using a particular motion (driven by the necessity to drive the stick through the mesh) to recover… food. The authors need to make a case to explain why they believe they actually witness any kind of ‘reinvention’.

'The paragraph added L449-454 does not address the fact that chimps may have some knowledge about the use of sticks to obtain food, for instance retrieving honey or jam from a termite mount, or from plastic tubes filled with food, which are quite often provided by zoos for enrichment. The authors have to state whether chimps had access to this kind of enrichment (through a list of stick behaviour they might be familiar with), and if that’s the case, how they can address this caveat. '

-L494-496: Once again, I am not convinced that the authors address the question of high fidelity social learning with their experiment, rather than the innovation abilities of the chimpanzees. I believe this distinction will have to be addressed in the reviewed version of this article.

Final comment on the discussion: I believe it might benefit from having subtitles or sections. At the moment, the reader may become a bit lost when reading the whole thing at once. Thus I would suggest re-organizing it logically and/or introduce section titles.

---

## Round 0.2 · Minor Revisions

The reviewer who recommended major revisions believes your paper now requires relatively minor revisions prior to publication, and I agree. Thank you for your careful attention to both reviewers' suggestions and comments. Assuming that you will handle these revisions as best you can, I will not send your ms. back out for re-review once you resubmit the final version. It seems likely that you and this reviewer will have to agree to disagree on some nontrivial issues. I look forward to receiving your final version and sending it off to press.

Reviewer 2 ·

Basic reporting

See below

Experimental design

See below

Validity of the findings

See below

Additional comments

I thank the authors for addressing (in details!) my comments. First, the article reads much better and it is much easier to follow the line of thoughts of the authors. While I still fundamentally disagree with the notion that chimpanzees were 'naive', as they had knowledge of sticks, I appreciate that the authors made an effort to pin down exactly what they are after, that is testing whether the form of a behaviour can be reinnovated individually.

I would probably request a few more things added in the intro and discussion (this, or I guess the reviewer-author exchange might be published together with the article so that it outlines the various opinions regarding the data...).

1) Regarding the intro, and looking at the comments, I still believe there is some disagreement about what is social learning and what is individual learning. When I wrote that even in humans,'down to the individual level, every behaviour learnt (including socially) depends on individual learning', my point was to stress out that each individual, to develop the behaviour, must learn it individually. In effect, an individual must have the physical and mental capacities to develop a given a behaviour to be able to develop them. I agree with the fact that individuals may not have the 'innovative abilities' to invent a new behaviour anew, but it does not make sense to write that 'many learnt behaviours are outside of the individual-LEARNING capabilities of each individual'. This would basically mean that you need to be several individuals to be able to acquire one particular behaviour (maybe this is the case for bird flocks, but not for humans, as far as I understand it). Rather, what the authors mean is that the behaviour we display now in 2017 would never have been innovated by someone who say lived in England in the XVIIIth century. But down to the individual, it's still their individual learning capacities that allow her/him to 'learn' a behaviour. To take the tango example given by the authors, while the moves of tango result from cumulative culture (so learning the moves needs social learning), the individual has to use individual-based learning mechanisms such as working memory to register and display the moves in the right order. So again, social learning is then socially-biased individual learning. See Heyes' arguments.

2) Second, regarding the discussion, I believe it would be worth discussing in the section about "not naive chimps?" the splitters vs lumpers argument, which the authors use to their advantage here. This gets back to my point that the authors demonstrate little more than chimps being good at transferring knowledge from one task to the other other, which contrasts with findings from the field. If one analyses each behaviour as a different innovation, then, clearly there will be many behaviours falling into the ZLS if an animal already knows a closely related behaviour (e.g. stick use for algae scooping). But that does not make a strong argument for ZLS as an hypothesis. In effect, that makes it weaker. In contrast, if an animal develops stone cracking from scratch (and without any human help), then it makes a much bigger case for the ZLS.


A few more minor comments:

Introduction:

-L129: A dot is missing.

Discussion:

-L419: I am not sure you should cite the Koops et al. 2014 paper here, if your point is to say that both opportunity and necessity interact to lead to the appearance of a tool behaviour. In contrast, Koops et al. 2014 state that only opportunity, but not necessity plays a role in the appearance of tool use. Rather, you should either cite the original book chapter (Fox et al. 1999, Intelligent tool use in wild Sumatran orangutans) or more recent papers that have advocated for such a view combining the two hypotheses (e.g. in capuchins: Moura & Lee, 2015 book chapter Necessity, unpredictability and opportunity: an exploration of ecological and social drivers of behavioural innovation; in chimps: Gruber et al. 2016 eLife).

-L441-442: I think the authors could mention here that innovation skills are in fact quite limited in young human children compared to their well demonstrated social learning skills (e.g. Nielsen 2013).

---

## Round 0.3 · accepted · Accept

Thank you for your thoughtful responses to reviewer suggestions and your patience with the review process.